# Multiverse Predictions for Habitability: Origin of Life Scenarios

McCullen Sandora [1,*], Vladimir Airapetian [2,3], Luke Barnes [4], Geraint F. Lewis [5] and Ileana Pérez-Rodríguez [6]

1   Blue Marble Space Institute of Science, Seattle, WA 98154, USA
2   Sellers Exoplanetary Environments Collaboration, NASA Goddard Space Flight Center, Greenbelt, MD 20771, USA
3   Department of Physics, American University, Washington, DC 20016, USA
4   School of Science, Western Sydney University, Locked Bag 1797, Penrith South DC, NSW 2751, Australia
5   Sydney Institute for Astronomy, School of Physics, A28, The University of Sydney, NSW 2006, Australia
6   Department of Earth and Environmental Science, University of Pennsylvania, Philadelphia, PA 19104, USA
*   Correspondence: mccullen@bmsis.org

**Abstract:** If the origin of life is rare and sensitive to the local conditions at the site of its emergence, then, using the principle of mediocrity within a multiverse framework, we may expect to find ourselves in a universe that is better than usual at creating these necessary conditions. We use this reasoning to investigate several origin of life scenarios to determine whether they are compatible with the multiverse, including the prebiotic soup scenario, hydrothermal vents, delivery of prebiotic material from impacts, and panspermia. We find that most of these scenarios induce a preference toward weaker-gravity universes, and that panspermia and scenarios involving solar radiation or large impacts as a disequilibrium source are disfavored. Additionally, we show that several hypothesized habitability criteria which are disfavored when the origin of life is not taken into account become compatible with the multiverse, and that the emergence of life and emergence of intelligence cannot both be sensitive to disequilibrium production conditions.

**Keywords:** multiverse; habitability; origin of life

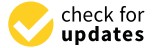



## 1. Introduction

### 1.1. Why Are We in This Universe?

The multiverse hypothesis states that universes aside from our own exist and may have different laws of physics [1,2]. This hypothesis is highly controversial among cosmologists because almost by their very nature, other universes are not directly observable, making it doubtful that we will ever be able to directly test this hypothesis [3]. This is part of a series of papers [4–10] that aims to make progress on this situation by providing a slew of indirect tests of the multiverse. The hope is that, even in the absence of direct observation, if it can be shown that the multiverse provides a framework for generating a litany of testable predictions, we will be able to collect evidence either for or against it.

The strategy we adopt is to use what is known as the *principle of mediocrity*, which states that our observations should be typical among the ensemble of all possible observers [11]. This is implemented in the multiverse framework by determining the probability that we would find ourselves in a universe such as ours, which we operationalize by computing the probability that we measure our particular values of the following fundamental (dimensionless) constants: the fine structure constant $\alpha$, which measures the strength of electromagnetism, the ratio of electron to proton mass $\beta = m_e/m_p$, the strength of gravity, as measured by the ratio of proton to Planck mass $\gamma = m_p/M_{pl}$, and the ratio of the up and down quark masses to the proton mass, $\delta_u = m_u/m_p$, $\delta_d = m_d/m_p$.

In this work, we compute the probability of measuring our observed values of these five constants given some habitability condition $\mathbb{H}$. For each variable $x$, we define this as

either the probability of being larger or smaller than our observed value $x_{obs}$, whichever is smaller—$\mathbb{P}(x_{obs}|\mathbb{H}) = \min(P(x < x_{obs}|\mathbb{H}), P(x > x_{obs}|\mathbb{H}))$. This penalizes both anomalously large and anomalously small parameter values. These are computed by defining a probability density function for observing any particular value of the constants, which is the product of two factors: $p(x) = p_{prior}(x)\,\mathbb{H}(x)$. Here, $p_{prior}(x)$ is the relative abundance of universes with a given set of values for the physical constants, and the habitability $\mathbb{H}(x)$ weights each universe by the number of observers it produces (defined deliberately vaguely here as a complex life form with human-esque intelligence). At the moment, we have a somewhat clear idea what the distribution of physical constants, based on generic arguments such as scale invariance that are, importantly, rather independent of the details of the underlying ultimate theory of physics [12], (though we explore the sensitivity of our analysis to this input in the Appendix A)[1]. However, due to the large uncertainties in the criteria for complex life, we cannot estimate the habitability of different universes with any degree of certainty.

It is precisely this lack of knowledge that enables us to make predictions, however. Instead of committing to a particular set of habitability criteria (such as whether life needs carbon, or plate tectonics, or to be around a sunlike star, etc.), we entertain various different criteria and compute the probabilities of our observations for each. As we have shown in our previous work, some criteria are drastically incompatible with the multiverse in the sense that they would make our presence in this universe highly unlikely, while adopting other habitability criteria makes our presence in this universe highly likely. We may then say that the multiverse framework favors the set of habitability criteria that make our observations likely. If we were to take the stance that the multiverse framework is true, this gives us concrete predictions for which habitability criteria are true and which are false. Though we do not know which habitability criteria are correct at the moment, we will eventually be able to determine this through a combination of better understanding the history of life on Earth and by finding other examples of life throughout the universe. When this knowledge is gained, we will be able to compare with the predictions the multiverse framework has given us; if the results match, we will have evidence for the multiverse, and if they do not, we will have evidence against it. Though this will not be an easy undertaking, it demonstrates that the multiverse framework is capable of producing concrete, testable predictions and as such deserves to be considered a scientific endeavor.

What remains is to incorporate as many habitability criteria into this framework as possible to obtain as many predictions for which are true as possible. In this paper, we consider the conditions for the origin of life as a potential determining factor for why we are in this universe. If the origin of life is indeed a difficult process, we then expect it to be sensitive to the local conditions of its emergence, and so, it could be reasoned that locales with suitable conditions are more likely to result in life. By this reasoning, universes which are more apt to produce locales suitable for the emergence of life would then be more habitable provided that they also allow for life to flourish. The conditions required for the origin of life, however, are unknown and divided into highly contrasting scenarios. Here, we consider several of these scenarios and determine which are compatible with our existence in this universe and which lead to the conclusion that other universes would be much more prolific at creating life.

### 1.2. What Is the Probability of the Emergence of Life?

We incorporate origin of life scenarios into this framework by expressing the probability of life emerging on a particular planet $p_{life}$ in terms of fundamental constants. This can then be incorporated into the total habitability of a universe through the Drake-style equation

$$\mathbb{H} = N_\star\, n_e\, p_{life}\, p_{int}\, N_{int} \tag{1}$$

where $N_\star$ is the total number of habitable stars in the universe, $n_e$ is the number of habitable planets per star, $p_{int}$ is the probability of developing intelligence on a planet with life, and $N_{int}$ is the number of observers created by planets that develop intelligence. These other

factors certainly play a large role in dictating the habitability of a universe but were the focus of our previous works and so will not be dealt with explicitly here. We will briefly allude to how these depend on the habitability criteria we consider in Section 6, where we provide references for further detail.

Equation (1) is a shorthand expression in the sense that many of these variables will depend on environmental conditions such as time in the universe's evolution, position within the galaxy, stellar/planetary mass, etc. This dependence will change depending on the habitability conditions being considered, but when such dependence occurs, it is to be integrated over the relative fraction for each variable, yielding a total quantity representing the full number of observers produced by a universe throughout its history. The full subtleties of this are not discussed here but are taken into account in our calculations; see especially [5] for further details, where we discuss the fraction of stars that reside in galaxies large enough to retain supernova ejecta, which are born late enough to be enriched in heavy elements and contain sufficient metallicity to form rocky planets but not so much as to produce hot Jupiters.

To standardize all origin of life scenarios used here, and in order to extrapolate their efficacy to universes with alternative values of the physical constants, we need a prescription for how likely the emergence of life is under each scenario. This prescription must be abstract enough to accommodate our limited understanding of both the processes involved and the details of other universes. For this, we take as a starting point that the probability of the emergence of life on a planet is proportional to the total amount of chemical disequilibrium present,

$$p_{\text{life}} \propto \Delta S. \tag{2}$$

Here, $\Delta S$ is a dimensionless quantity, equivalent to the number of molecules produced by a given mechanism that are out of chemical equilibrium with the environment and therefore available for prebiotic chemical processes to participate in rudimentary chemical reaction networks. These disequilibrium molecules then provide the feedstock that over time may ultimately develop into a system that could be construed as life.

We believe this is a reasonable ansatz to start with, as it represents the total amount of information able to be processed by the entire planetary system. Obviously, other factors aside from just total disequilibrium come into play. Many other aspects of environmental conditions may be equally important to the emergence of life, including concentrations, gradients, cycles, and phases of matter. It may well be that no matter how much disequilibrium is present in interstellar space or on planets with no liquid phase, life can never develop, and therefore, our utilization of this simplified formula should be taken in conjunction with other habitability conditions. To ensure these additional constraints are met, in this paper, we involve this disequilibrium ansatz into a formalism that optionally includes twelve other potential habitability conditions.

Though our disequilibrium ansatz will be the default used throughout most of this paper, we will discuss well-motivated alternatives for particular scenarios as the need arises, and so here we briefly outline some objections one may have to this formula, as well as some extensions.

Perhaps the most obvious objection to the above is that, taken blindly, it implies that if one were to arrange a scenario where the total disequilibrium production were sufficiently large, the probability of developing life would exceed 1. Naturally, we expect this formula to break down well before this point, asymptoting instead to some constant value. A candidate extension to our ansatz that takes this into consideration would be $p_{\text{life}}(\Delta S) = p_0(1 - exp(-\Delta S/S_0))$. This reduces to our original form in the limit $\Delta S \ll S_0$ and asymptotes to the value, $p_0$, which may be potentially much less than 1. We make no guesses as to what the values of $S_0$ and $p_0$ may be.

It should be noted that this formula represents the interpolation between two regimes, the first of which is $\Delta S \ll S_0$, in which the emergence of life is sensitive to the details of the processes involved, and the secondof which is $\Delta S \gg S_0$, in which it is not. This latter regime may be regarded as a null hypothesis, wherein the factor $p_{\text{life}}$ does not play any role

in determining our location throughout the universe or multiverse. As such, consideration of this full equation may be foregone in favor of only considering the two limiting regimes. This raises an important corollary of our ansatz; it indicates that, if life is sensitive to the amount of disequilibrium produced, the probability of life emerging is necessarily small. Therefore, determining the viability of this ansatz allows us to probe the expectation for the overall abundance of life throughout our universe.

Lastly, we note that even this extended treatment cannot be fully accurate, as beyond a certain point, contributing more disequilibrium will adversely affect the chances of life's emergence. Our ansatz makes no reference to potentially damaging combustion reactions that may lead to the breakdown of prebiotic material if disequilibrium exceeds a certain threshold. Beyond that, material in enough abundance could drastically impact the overall planetary dynamics far beyond creating localized conditions for prebiotic chemistry (for instance, several hundred bars of organic material would severely alter the dynamics of a planet's atmosphere-hydrosphere system).

A second potential objection is that one might expect the probability of emergence to not scale exactly linearly with disequilibrium, but rather as some other power, $p_{\text{life}} \propto \Delta S^n$. This conclusion may be reached from a large variety of considerations, all relevant to different origin of life scenarios. For instance, if one imagines life to have arisen from a loosely connected network of localized prebiotic reaction sites, as would have occurred in hydrothermal vents, subaerial lakes, coastlines, etc., one would suspect the exponent $n$ to be somewhere between 1 and 2, dependent on the degree of connectivity between the different sites. Likewise, if there are any loss processes that are independent of reservoir size, as may be the case for UV loss of a reducing atmosphere, then the total system lifetime would be proportional to the amount of disequilibrium, and $n$ would be close to 2. Thirdly, many chemical reactions are proportional to the products of reactant densities, any of which may scale linearly or nonlinearly with total disequilibrium. Lastly, one may regard the origin of life as a 'hard step' process, whereby each step in a chain of necessary innovations is proportional to disequilibrium, leading to the concatenation of all processes being proportional to some integer [14]. In light of all these potential rationales, we comment on this generalization of our initial ansatz where appropriate.

Another potential deficiency of our formula is that it does not take any temporal component into consideration, treating as equally likely the immediate one-time dumping of an amount of disequilibrium $\Delta S$ or the continued meting out of the same amount over geological time. In a more detailed setup, where other processes may come into play, there would most likely be a difference in these two scenarios.

As a simplest example, imagine a setup where the reservoir of initial disequilibrium molecules is lost to two processes: the participation in prebiotic chemistry, so as to make organic molecules, and some abiotic process, such as deposition, reaction with some sterile chemical species, or loss to space. In many scenarios, we may regard both of these loss processes as proportional to concentration $n$, leading to the following evolution:

$$\dot{n} = -\left(k_{\text{prebio}} + k_{\text{loss}}\right)n \tag{3}$$

where the $k_i$ are reaction rates. One can show that in such a setup, the total amount of material available for prebiotic chemistry is then

$$p_{\text{life}} \propto \frac{k_{\text{prebio}}}{k_{\text{prebio}} + k_{\text{loss}}} \Delta S \tag{4}$$

When the prebiotic loss rate is much greater than the abiotic, this reduces to our formula from before. When the abiotic loss is nontrivial, however, the total amount of available material may be substantially smaller than what is initially present. This additional factor may lead to additional parameter dependence, as both rates will generically scale with constants in different ways.

One may concoct any number of other alternatives to the above extensions, but these will be sufficient for a reasonably thorough investigation into the effects different ansatzes will have on our formalism while providing a sufficient level of realism to capture many of the effects one could worry about.

Below, we calculate the disequilibrium generated in several origin of life scenarios. In Section 2, we begin with the classic Miller–Urey prebiotic soup setup, taking lightning, solar energetic protons, and ultraviolet light as energy sources. In Section 3, we consider hydrothermal vents as the origination sites of life, with oxidation-reduction reactions acting as chemical energy sources. In Section 4, we consider impacts as the dominant source of prebiotic material and consider four variants: the case where prebiotic molecules are initially present on comets, the case where prebiotic material is created from shock synthesis during impact, with either small comets or asteroids as the dominant source, and the case where disequilibrium is created by a single large impactor. In Section 5, we consider the panspermia scenario for the origin of life and discuss both the interplanetary and interstellar variants. Lastly, in Section 6, we fold in the values of the disequilibrium we find for each scenario into calculating the probabilities of our observations.

Roughly speaking, if the disequilibrium production for an origin of life scenario is much greater in other universes, this would discount that scenario on the grounds that we would have been much more likely to have arisen in a universe more conducive to life's emergence. Likewise, if disequilibrium production is larger than usual in our universe for a scenario, that scenario is favored within the multiverse framework. This logic is formalized when we consider the probabilities of our observations, where we find strong evidence against both types of panspermia scenarios and weaker evidence against the solar energetic proton scenario. The other scenarios we consider are neither strongly favored nor disfavored by the multiverse framework. We also note that several previously considered habitability criteria which were untenable with the null hypothesis become consistent with the multiverse when taking certain origin of life scenarios.

## 2. Prebiotic Soup

### 2.1. Lightning Rate

In the classic prebiotic soup origin of life scenario, organic compounds are produced in a reducing environment with the addition of a free energy source capable of breaking molecular bonds [15]. The source of energy was originally conceived to be lightning flashes. Though this scenario works best for a more reducing atmosphere than is realistic for early Earth, it has been shown to be somewhat effective in neutral atmospheres as well [16] and may have taken place where some local atmospheric conditions achieve sufficient reducing power [17].

To determine the amount of disequilibrium produced in this scenario, we need to estimate the total amount of energy produced by lightning flashes over the course of a planet's evolution. Because the number of bonds broken by any lightning bolt is much smaller than the total atmospheric reservoir, production via this mechanism will always be close to the theoretical maximum. As proposed in [18], the most predictive quantity for local lightning power on Earth is

$$P_{\text{lightning}} = \epsilon_{\text{lightning}} \, P_{\text{rain}} \, \frac{E_{\text{conv}}}{\mu \, m_p} \tag{5}$$

Here, $\epsilon_{\text{lightning}}$ is an efficiency factor, found to be $\approx 0.01$ in the above reference; $P_{\text{rain}}$ is the precipitation rate; $E_{\text{conv}} \sim T_{\text{atm}}$ is the convective energy in the atmosphere; and $\mu$ is the dimensionless mean atmospheric molecular weight. This expression can be applied to the planet as a whole to derive the total lightning rate, and the dependence on physical constants can be identified.

The global precipitation rate can be given simply by the power received from the sun divided by the evaporation energy required to evaporate one water molecule,

$$P_{\text{rain}} \sim S_0 \, A_{\text{ocean}} \frac{m_{H_2O}}{E_{H_2O}} \tag{6}$$

Here, $S_0$ is the solar constant, or more generally the amount of incident radiation on a terrestrial planet's surface; $A_{\text{ocean}} \sim A_{\text{terr}}$ is the area of surface liquid water on a planet; and $E_{H_2O}$ is the amount of energy required to break the molecular bonds of a water molecule. The presumption with this equation is that a substantial fraction of photons incident on a planet's exposed water area will go into evaporation. This simple expression, when applied to the Earth, yields an estimate remarkably close to the observed value of $10^{15}$ kg $H_2O$/day [19]. When this is used for the lightning rate, if we restrict our attention to temperate planets, where $T_{\text{atm}} \sim E_{\text{mol}}$, the mass-energy conversion exactly cancels the factor $E_{\text{conv}}$. Then, we arrive at the exceedingly simple expression

$$P_{\text{lightning}} = \epsilon_{\text{lightning}} \, S_0 \, A_{\text{terr}} \tag{7}$$

The efficiency factor $\epsilon_{\text{lightning}}$ will depend on the temperature and atmospheric water mixing ratio [20], but dimensionally, lightning power is proportional to the power received by the sun. This expression also neglects lightning generated by geothermal processes, which may have been substantially greater on early Earth [21].

We now need to determine the amount of disequilibrium produced given this power. The amount of nitrogen fixed by lightning depends mildly on the oxidation state of the atmosphere [22,23], but generically is close to the maximum efficiency. In [21], lightning was found to produce 1–9 × $10^{16}$ molecules NO/J, with maximally efficient production being $6.2 \times 10^{17}$ molecules NO/J. It is important to note that fixation results not from the direct destruction of $N_2$ molecules but instead through the destruction of less energetically bound intermediaries such as $CO_2$, which then reacts with $N_2$ [24]. This makes disequilibrium production less sensitive to atmospheric composition.

The total disequilibrium due to lightning power will be given by the disequilibrium production rate multiplied by the total time that the planetary system evolves. For this, we take the time a planet remains in the temperate zone, which is a substantial fraction of the stellar lifetime $t_\star$. Then, we have

$$\Delta S_{\text{lightning}} = \tilde{\epsilon}_{\text{lightning}} \, S_0 \, A_{\text{terr}} \, t_\star \tag{8}$$

Here, we have defined $\tilde{\epsilon}_{\text{lightning}} \approx 10^{-4}$ as the product of the various efficiency factors used throughout. We use expressions for the instellation, area and lifetime of temperate, terrestrial planets[2] found in [4] to express this in terms of physical constants as

$$\Delta S_{\text{lightning}} = 1.1 \times 10^{-5} \, \tilde{\epsilon}_{\text{lightning}} \frac{\alpha^7 \, \beta^{3/2}}{\lambda^{5/2} \, \gamma^4} \tag{9}$$

Here and in all following expressions of disequilibrium, we normalize this to the value of $3 \times 10^{45}$ NO molecules as a representative estimate. It should be noted that the actual normalization is unimportant for our purposes; this drops out when computing the probability of our observations, which only depends on the likelihood of life emerging in our universe relative to universes with differing physical constants. We also keep track of how this depends on stellar mass through the dimensionless parameter $\lambda = M_\star/M_{\text{Ch}}$, with the Chandrasekhar mass $M_{\text{Ch}} = 122.4 M_{pl}^3/m_p^2$.

## 2.2. Solar Energetic Protons (SEPs)

An additional source of chemical disequilibrium in the prebiotic atmosphere is given by the incidence of high-energy protons blown off by the sun. These have been shown to

be more productive in creating chemical disequilibrium than lightning in more realistic pH conditions [25]. Estimates for the total disequilibrium produced by this source must take into account not only the total number of protons incident on the planet but also the fraction that make it to the lower atmosphere, which is necessary for the energy to be imparted in a potentially productive location [26].

The total disequilibrium produced in this scenario is estimated as

$$\Delta S_{\text{SEP}} = \frac{\dot{M}_{\text{wind}}\, v^2_{\text{wind}}}{2\, E_{\text{Rydberg}}} \frac{R^2_{\text{terr}}}{4\, a^2_{\text{temp}}} f_{\text{trop}}\, t_{\text{brake}} \tag{10}$$

Here, $\dot{M}_{\text{wind}}$ and $v_{\text{wind}}$ are the rate and speed of stellar wind, given in terms of physical constants in [10]. This neglects flux occurring during transient coronal mass ejection events, which represent a subdominant contribution to the total mass flux [27]. However, we do not expect this contribution to alter the overall scaling with constants. The relevant timescale in this scenario is $t_{\text{brake}}$, the spin-down time which dictates a star's transient violent phase, as most of the particle flux occurs during this initial phase [28]. This is also estimated in [10] and is an algebraic expression in the physical constants, with the exception that it is inversely proportional to the fraction of stellar field lines, which are open $f_{\text{open}}$. This factor has a more complicated dependence on constants, which we do not display here.

The fraction of energetic protons that reach the troposphere $f_{\text{trop}}$ can be determined by first noting the energy required for a particle to make it through the atmosphere given the standard stopping formula [29]:

$$\frac{dE}{dx} = -8\pi\, \frac{a^2_{\text{Bohr}}\, E^2_{\text{Rydberg}}}{E}\, n(x) \tag{11}$$

This simplified form is used, where we neglect logarithmic and form factor corrections, and is valid for energies above 500 eV and is not sensitive to atmospheric composition. With this, and an exponential density profile $n(x) = n_0 \exp(-x/h_{\text{atm}})$, we find the minimum energy required for a proton to make it through the atmosphere is

$$E_{\text{min}} = \sqrt{16\pi\, a^2_{\text{Bohr}}\, n_0\, h_{\text{atm}}}\, E_{\text{Rydberg}} \tag{12}$$

The fraction of particles exceeding this energy is obtained through the cumulative distribution of particle energies, which takes the form $c(E) = (E_{\text{wind}}/E)^{k_{\text{SEP}}}$. Though the exponent $k_{\text{SEP}}$ depends on flare energy, in the main text, we use $k_{\text{SEP}} = 1$ [30] and explore the impact of this parameter choice in the Appendix A. The typical (minimum) particle energy is set by the escape velocity of the sun, $E_{\text{wind}} \sim G\, M_\star\, m_p/R_\star$, where $G$ is Newton's gravitational constant.

With this, the total disequilibrium produced by solar energetic protons is

$$\Delta S_{\text{SEP}} = 0.053\, \frac{\alpha^{19/2}\, \beta^{3/2}}{f_{\text{open}}\lambda^{69/20}\, \gamma^4} \left( \frac{\alpha^{5/2}\, \gamma^{1/3}}{\lambda^{17/20}\, \beta^{1/12}} \right)^{k_{\text{SEP}}} \tag{13}$$

Note that these expressions take the influence of the planet's magnetic field to not affect the overall flux. Indeed, while magnetic fields do shield most of the planet, they do so by funneling particles toward the poles, which will not have a drastic impact on the total disequilibrium produced [31]. Further elaboration on this was explored in [10]. We have also taken the atmospheric density to be set by material delivered during accretion, the total mass of which is determined in [10]. Considering additional sources of atmosphere, such as nitrogen in initial planetesimals, does not alter the dependence on constants by an appreciable amount.

*2.3. Ultraviolet Light (XUV)*

Additionally, we may consider stellar UV and X-rays (collectively called XUV) as sources of disequilibrium. However, the estimation of the total disequilibrium produced by these methods is complicated by the fact that photoreactive atmospheric content may act as a highly effective shield, limiting the disequilibrium produced in the lower atmosphere. Though ozone plays the role of shield on present-day Earth, its development is more generic and can be due to a number of potential compounds, including hazes, nitrogen-bearing molecules [32], and sulfur-oxygen compounds [33]. We neglect potential shielding in our estimates of incident XUV radiation and simply take

$$\Delta S_{\text{XUV}} = \frac{L_{\text{XUV}}}{E_{\text{Rydberg}}} \frac{R_{\text{terr}}^2}{4\, a_{\text{temp}}^2}\, t_{\text{brake}} \tag{14}$$

Using expressions for the XUV luminosity and spin-down time from [10], we find

$$\Delta S_{\text{XUV}} = 3.9 \times 10^{-8}\, \frac{\alpha^{17/2}\, \beta^{3/2}}{f_{\text{open}}\, \lambda^{23/20}\, \gamma^4} \tag{15}$$

## 3. Hydrothermal Vents

Hydrothermal vents are regions on the seafloor generated from the interaction of material from the deep subsurface with seawater, which releases hydrothermal fluid. Nearly since their discovery, they have been discussed as an attractive source for the origin of life due to their nature as a source of disequilibrium, rich prebiotic chemistry, and temperature and pH gradients [34]. This view is bolstered by the fact that the reducing environments typical of hydrothermal vents are conducive to amino acid and peptide synthesis, and many reactions that occur there show a resemblance to what are inferred to be some of the oldest metabolic pathways, such as the acetyl-CoA cycle [35]. Hydrothermal vents are natural locations for the hypothesized iron-sulfur world scenario for the origin of life [36]. This scenario is attractive because the pyrite potential can readily reduce $CO_2$ directly, which provides material for prebiotic reactions [37].

However, significant obstacles to the origin of life at hydrothermal vents remain. The oligomerization of both amino and nucleic acids requires dehydration conditions [38], and it has proved challenging to obtain amino acid polymers of significant length with the temperature and pH conditions typical of hydrothermal vents [39].

The hydrothermal venting associated with white smokers, in particular, has recently emerged as an attractive site for the origin of life due to their longevity and higher pH [40], which has been shown to be better for vesicle formation [41]. However, this poses a challenge to RNA world origin of life theories, as RNA is unstable in alkaline conditions [42]. The distinction between the two types of vents is not important for our purposes.

If hydrothermal vents were indeed the locations of the origin of life, this would have strong implications for the expected distribution of life throughout the universe. Life could then potentially arise on any tectonically active body with a liquid ocean in contact with the sea floor. As several outer moons of our solar system are suspected to have hydrothermal vent systems in their subsurface oceans [43] in potentially much greater abundance [44], it raises the possibility that life may arise in these locales as well. However, as pointed out in [45], the amount of energy produced on smaller bodies may be orders of magnitude smaller than on Earth. If the probability of life's emergence scales with disequilibrium production, as per our ansatz, this would entail a correspondingly decreased probability of life emerging on these bodies.

We must now estimate the total amount of disequilibrium generated by the hydrothermal vents. As discussed in [46], the total amount of degassed volatiles $\Gamma_{\text{vent}}$ can be de-

termined by the seafloor spreading rate $v_{\text{spread}}$, the melt generation depth $d_{\text{melt}}$, and the mantle volatile concentration $f_{\text{vol}}$ as

$$\Gamma_{\text{vent}} = f_{\text{vol}} \, 2\pi \, R_{\text{terr}} \, v_{\text{spread}} \, d_{\text{melt}} \tag{16}$$

We will discuss the physical factors $v_{\text{spread}}$ and $d_{\text{melt}}$ first, and then return to the compositional factor $f_{\text{vol}}$.

The main disequilibrium reaction is serpentization, which stems from the fact that mantle conditions are highly reducing with respect to surface conditions, leading iron to form bonds with oxygen that are unstable at the surface [47]. Chemically, the reaction can be summarized as $2\text{FeO} + \text{H}_2\text{O} \rightarrow \text{Fe}_2\text{O}_3 + \text{H}_2$, and results in $10^{12}$ mol $\text{H}_2$/year [48]. Over the course of Earth's history, this has resulted in $\Delta S_{\text{vent}} \approx 3 \times 10^{45}$ disequilibrium chemical bonds available for biochemical reactions to utilize.

The seafloor spreading speed is set by advection due to heat transfer, $v_{\text{spread}} \sim Q/(4\pi R_{\text{terr}}^2 n \Delta T)$ [49]. A worthwhile understanding of the magnitude of $v_{\text{spread}}$ can be obtained if we substitute the expression $Q \sim G M_{\text{terr}} \rho \kappa_{\text{heat}}$, where $\kappa_{\text{heat}}$ and $\rho$ are the thermal diffusivity and density of rock. With this, we find $v_{\text{spread}} \sim \kappa_{\text{heat}}/R_{\text{terr}}$, where we have used the definition of a terrestrial planet, $G M_{\text{terr}} m_p / R_{\text{terr}} \sim E_{\text{vib}} \sim \Delta T$. If we continue and use $\kappa_{\text{heat}} \sim c_s a_{\text{Bohr}}$ and $R_{\text{terr}} \sim c_s/\sqrt{G\rho} \sim c_s t_{\text{day}}$, where $c_s \sim \sqrt{E_{\text{vib}}/m_p}$ is the speed of sound in rock [6], we find the simple expression

$$v_{\text{spread}} \sim \mathcal{O}(100) \frac{a_{\text{Bohr}}}{t_{\text{day}}} = 35.2 \sqrt{\alpha \, \beta} \, \gamma \tag{17}$$

This makes the rates of processes underlying mantle convection and plate tectonics conceptually simple, as atomic defects in the Earth's internal rock structure must move by several (hundred) locations per day in order to be operational.

Lest this order of magnitude derivation be over-interpreted, we hasten to point out several things: in several steps of our derivation, we have used conditions that hold only for terrestrial planets, and so we do not expect it to hold for other types of planets or moons. In our simplified expression for heat flow, we have neglected all time dependence, which is necessary to account for the fact that the spreading rate has changed throughout Earth's history. Our expression does not implicate that daily cycling induces the dominant defect-moving forces within the mantle, but rather that as a consequence of the planet being terrestrial, the strength of internal convective stresses is the same order of magnitude as the gravitationally set orbital forces. Lastly, in our derivation, we have conflated $t_{\text{day}} \sim t_{\text{breakup}} = 1/\sqrt{G\rho}$ because this sets the natural rotational speed for rocky planets. Thus, we would not be able to use this reasoning to conclude that plate tectonics should be linearly related to rotational speed, nor that it should be nearly absent on tidally locked planets.

Next, we must determine what sets the melt depth $d_{\text{melt}}$. This is given by the depth at which the ascending mantle meets the eutectic point of basalt, triggering the mantle to melt, and is approximately 40 km for Earth [46]. Its depth may be determined by finding the point where the mantle temperature equals the solidus temperature. From [50], the solidus temperature depends on pressure through a rather complicated expression, but for shallow depths, the dependence is well approximated by the linear equation $T_{\text{sol}} = P/n + T_{\text{melt}}$, where $P$ and $n$ are the pressure and number density, and $T_{\text{melt}} = 1100$ K. Because this only leads to about a degree increase in solidus temperature per km, this may be well approximated by the melting temperature at the surface.

We use the expression for temperature profile with depth

$$T(r) = T_{\text{surface}} + (T_{\text{mantle}} - T_{\text{surface}}) \, \text{erf}\left(\frac{d}{2\sqrt{\kappa_{\text{heat}} t_{\text{floor}}}}\right) \tag{18}$$

Here, $\sqrt{\kappa_{\mathrm{heat}} t_{\mathrm{floor}}}$ is the lithosphere thickness [51]. Since this is the only relevant distance scale in these equations, it also sets the melt depth, $d_{\mathrm{melt}} \sim \sqrt{\kappa_{\mathrm{heat}} t_{\mathrm{floor}}}$. Lastly, the average seafloor age is simply given by $t_{\mathrm{floor}} \sim R_{\mathrm{terr}} / v_{\mathrm{spread}}$.

One may worry that, since the solidus temperature increases with depth, while the mantle temperature eventually asymptotes, there may be parameter values for which the mantle temperature never exceeds the melting point, and no melting occurs. Roughly, the condition for this is $m_p g d_{\mathrm{melt}} / T_{\mathrm{mantle}} \gtrsim 1$, where $g = G M_{\mathrm{terr}} / R_{\mathrm{terr}}^2$ is the surface gravity. However, it can be shown that this ratio is actually independent of constants for terrestrial planets, and so melting in the crust occurs throughout the multiverse. This follows from the terrestriality condition, as well as noting that melt depth can be rewritten as $d_{\mathrm{melt}} \sim \sqrt{E_{\mathrm{vib}} R_{\mathrm{terr}} / (g \, m_p)} \approx 0.006 R_{\mathrm{terr}}$. Thus, we also expect the ratio $d_{\mathrm{melt}} / R_{\mathrm{terr}}$ to remain roughly constant throughout the multiverse.

These expressions lead to a total amount of disequilibrium production, if the planetary lifetime is set by the stellar lifetime, of

$$\Delta S_{\mathrm{vent}} = 66.5 \, f_{\mathrm{vol}} \frac{\alpha^{9/2}}{\lambda^{5/2} \, \gamma^3} \tag{19}$$

The other factor dictating the outgassing rate, $f_{\mathrm{vol}}$, is set by the compositional conditions of the Earth. Since serpentization is the main source of disequilibrium production [40], if mantle iron is much less abundant, disequilibrium production will correspondingly be less. Lack of mantle iron abundance could occur for two separate reasons: firstly, if iron were absent initially, and secondly, if stratification were to proceed differently. In our previous paper [8], we found conditions on the physical constants that are required for iron to be the endpoint of stellar nucleosynthesis. We may robustly expect a nucleus with atomic weight 56 to be the ultimate product of fusion, as this possesses a doubly magic number of nucleons. However, in stellar fusion, nickel-56 is initially produced, which thereafter beta decays until it becomes a stable element; which element this is is dictated by nuclear binding energies. If

$$E_{Fe}^{56} - E_{\mathrm{Co}}^{56} > 0 \tag{20}$$

iron-56 will be stable; otherwise, it would decay to manganese-56. Likewise,

$$E_{Fe}^{56} - E_{\mathrm{Mn}}^{56} > 0 \tag{21}$$

is required for cobalt-56 to be unstable. These put limits on the physical constants, beyond which iron abundance would be significantly diminished.

A further requirement is that the planet should not become fully stratified, as this would result in the near-complete sequestration of iron in the core. In [52], it was found that iron will remain in the mantle only if $(\mathrm{Mg} + 2\mathrm{Si})/\mathrm{O} < 1$. This sets the condition for oxygen unpaired with silicon or magnesium to be present in the mantle, which is a requisite for FeO. In [10], we found that this equates to a condition on the Hoyle resonance energy $E_R = 0.626(m_u + m_d) + (0.58\alpha - 0.0042)m_p > -0.874 \, \mathrm{keV}$ based off the element abundance calculations of [53]. Therefore, we can take the mantle volatile abundance to be

$$f_{\mathrm{vol}} = 0.004 \, \theta\!\left(E_{Fe}^{56} - E_{\mathrm{Co}}^{56}\right) \theta\!\left(E_{Fe}^{56} - E_{\mathrm{Mn}}^{56}\right) \theta(E_R + 0.874 \, \mathrm{keV}) \tag{22}$$

where the numeric prefactor is set according to [54].

One may also worry about the sulfur mantle abundance, as the formation of pyrite, $\mathrm{FeS} + \mathrm{H}_s\mathrm{S} \rightarrow \mathrm{FeS}_2 + \mathrm{H}_2$, is a secondary disequilibrium process. This is especially relevant for the iron-sulfur world, which heavily relies on the presence of sulfur for the emergence of life [55]. Again in [8], we found conditions on the fundamental parameters

$$f_{\mathrm{vol}}^{(S)} = \theta(E_S^{32} - E_P^{32}) + \theta(E_S^{31} - E_P^{31}) + \theta(E_{Cl}^{35} - E_S^{35}) \tag{23}$$

This may be incorporated into $f_{\rm vol}$ as an additional factor, but does not appreciably affect the resulting computation.

## 4. Exogenous Delivery

A third class of scenarios for the source of organic material on early Earth is via impacts. Meteors have high organic content and have been shown to host extremely high chemical diversity [56]. Due to the uncertainties in the details of this pathway, we consider the following variants: (i) the delivery of organic material via interplanetary dust particles, (ii) shock synthesis via (iia) comets and (iib) asteroids, and (iii) substantial atmospheric reduction through a single large impact.

### 4.1. Interplanetary Dust Particles (IDPs)

As outlined in [57], the delivery of organics via interplanetary dust particles may have represented the dominant source of organic material on early Earth. In this scenario, the vast majority (99.9%) of cometary material was delivered in the initial transient phase of planet system formation, most of which was in the form of IDPs. After this, the cometary population and injection rate into the inner system stabilized to the rate we currently observe. As such, the total amount of organic material is set by the total amount of initial cometary material $M_{\rm comets}$ multiplied by the fraction which is in organics $\epsilon_o$ and the fraction which hit Earth $f_{\rm hit}$.

$$\Delta S_{\rm IDP} = \epsilon_o \, f_{\rm hit} \, \frac{M_{\rm comets}}{m_p} \qquad (24)$$

In [58], interstellar dust particles were reported to be several percent organics by mass, a fraction which we do not expect to vary much for other parameter values. As described in more detail in [59], the fraction of cometary material that hit Earth is given by the ratio of gravitational cross-sections of the Earth and the Sun, $f_{\rm hit} = R_{\rm terr}^2/(2R_\star a_{\rm temp})$. The total amount of material is set by the initial disk mass, which we take to be proportional to stellar mass. An expression for the total cometary mass incident on a planet in terms of physical constants is found in [9].

Another important aspect of this delivery scenario is that organic compounds must have been created in appreciable amounts on cometary grains. Indeed, amino acids have been created in analogue cometary conditions [60], found in sample returns from the Stardust mission [61], and ribose [62] and nucelobases [63] have been found in meteorites. Since the ultimate relevance of these findings to the origin of life is unknown, it is informative from a multiverse perspective to wonder whether the ubiquity of complex organic molecules in interplanetary space is a generic feature of universes, or somehow special to ours. To investigate this, we use the formalism developed in [64], who show on generic grounds that the typical (and maximum) molecular size produced from the evolution of a chemical system scales as $\sigma_{\rm chem} \sim \sqrt{D_{\rm chem}\,t}$, where $D_{\rm chem}$ is a coefficient dictating the rate of diffusion through molecular configuration space, and $t$ is the total duration of the evolution. The diffusion coefficient is proportional to incident radiation as

$$D_{\rm chem} = \epsilon_{\rm mol} \, \frac{\Phi_{\rm XUV} \, A_{\rm molecule}}{\langle E_{\rm XUV} \rangle} \qquad (25)$$

where $\Phi_{\rm XUV}$ is the flux of photons capable of exciting electronic transitions. As per [65], we take the main source of chemical disequilibrium in the protoplanetary nebula to be from the parent star and in the X-ray window. The molecular area is given as a multiple of the square of the Bohr radius, $A_{\rm molecule} \sim \pi a_{\rm Bohr}^2$, the typical X-ray energy is set by the Rydberg $\langle E_{\rm XUV} \rangle \sim E_{\rm Rydberg}$, and $\epsilon_{\rm mol}$ is an efficiency factor, found to be $\epsilon_{\rm mol} \approx 10^{-4}$ in [60]. Note that, with this, the chemical evolution is set by the fluence (time-integrated flux) in accordance with the Bunsen–Roscoe law. To finally set the dynamics, we take the X-ray flux and duration of a young star from [10] and normalize the value of the average molecular weight

to be a few hundred (660) Daltons, as is appropriate for the Murchison meteorite. With this, we find

$$\sigma_{\text{chem}} = 6.9 \times 10^{-5} \, \frac{\alpha^{35/8} \, \beta^{31/24}}{\lambda^{79/96} \, \gamma^{53/48}} \tag{26}$$

In order for organic material to be present on IDPs, we enforce that this value should be larger than the molecular weight of the smallest amino acid glycine, $\mu_{\text{min}} = 75$ Daltons. This places a boundary in parameter space, beyond which large prebiotic molecules do not occur in space. This is most sensitive to the fine structure constant; we find that, with the other constants held fixed, if $\alpha$ were decreased below 61% of its observed value, no amino acids would be present in cometary dust.

The total disequilibrium produced in this scenario is then

$$\Delta S_{\text{IDP}} = 1.2 \times 10^{-5} \, \frac{\alpha^6 \, \beta^{1/2}}{\lambda^{31/20} \, \gamma^{7/2}} \, \theta(\sigma_{\text{chem}} - \mu_{\text{min}}) \tag{27}$$

*4.2. Impact Synthesis*

The second variant of the delivery scenario is that the bulk of organic material was not delivered to Earth but was created during the initial impact from reduced material present on the impactor. For objects less than $\approx 100$ m (set by comparing the mass of the object with the mass of the atmospheric column it traverses, assuming that a significant fraction of molecular collisions results in the breaking of a molecular bond), the impactor is disintegrated in the atmosphere. Organic material is created in these conditions at fairly high yields, and it was argued in [66] that this process is the dominant source of amino acids. (However, it was argued in [67] that shock synthesis of formaldehyde is subdominant to a subsequent, prolonged phase of photochemical synthesis.) To estimate the total amount of disequilibrium created via this process, the previous expression must be modified to compare the total energy delivered by material to the energy needed to break an atomic bond:

$$\Delta S_{\text{impacts}} = \epsilon_o \, f_{\text{hit}} \, \frac{M_{\text{impactors}} \, v_i^2}{2 \, E_{\text{mol}}} \tag{28}$$

Here, another difference between comets and asteroids arises. For comets, the impactor speed is on the order of the orbital speed of temperate planets, as cometary material is likely to impact from any direction. Conversely, asteroidal material is likely to be corotating with the Earth, and so the impactor speed is set by the typical eccentricity times the orbital speed, or the escape velocity of the planet, whichever is greater. Here, for simplicity, we focus on the latter. Comparing Equations (24) and (28) allows us to see why there is debate about which process should be dominant. Since the two expressions differ only by a factor $v_i^2 \sim E_{\text{mol}}/m_p$, and this factor is roughly 1 for terrestrial planets, the material created during shock synthesis is roughly equal to the amount delivered. For larger planets, shock synthesis dominates, while for smaller planets, delivered material dominates.

With this, the disequilibrium produced by comets is

$$\Delta S_{\text{comets}} = 5.4 \times 10^{-7} \, \frac{\alpha^9 \, \beta}{\lambda^{23/10} \, \gamma^4} \tag{29}$$

and the disequilibrium produced if asteroids are the main source, as argued to represent the main source owing to a period of gas giant instability in [68], is

$$\Delta S_{\text{asteroids}} = 0.94 \, \frac{\kappa \, \lambda^{21/10}}{\alpha^4 \, \beta^{4/3} \, \gamma^{8/3}} \tag{30}$$

We have used the asteroid mass determined in [9], and utilize the parameter $\kappa = 1.1 \times 10^{-16}$, which parameterizes galactic density.

### 4.3. Single Large Impact (Moneta)

The third variant of the delivery scenario is that of a single large impactor, dubbed Moneta. This scenario is motivated by the late veneer, whereby a substantial amount of material was delivered to Earth after planetary differentiation [69]. The fact that Earth seems to be enriched in crustal siderophile elements, while the moon does not, indicates that these were delivered in a single impact, as argued in [70]. This single impactor could have delivered enough reduced iron to substantially reduce the atmosphere for an extended period of time, resulting in temporarily more productive atmospheric chemistry [71]. Detractors of this scenario note, however, that the majority of iron during such an impact is deposited in the Earth's interior, limiting the reducing potential of this scenario [72].

Large impacts are stochastic by nature, but the overall scale of a typical impacting body is set by the isolation mass of the system, $M_{iso} = (2\pi \Sigma a^2)^{3/2}/M_\star^{1/2}$, where $\Sigma$ is the disk surface density and $a$ is semi-major axis. Therefore, if a sizable fraction of the material is involved in the production of an $H_2$ atmosphere, the total disequilibrium produced can be simply expressed as

$$\Delta S_{moneta} = 940.6 \, \frac{\kappa^{3/2} \, \lambda^{25/8}}{\alpha^{15/2} \, \beta^3 \, \gamma^{9/4}} \tag{31}$$

However, in this case, we expect a deviation from the formula $p_{life} \propto \Delta S$, as most of the hydrogen in this transient atmosphere will be lost to space before it has a chance to become incorporated into organic molecules. To account for this, we use the generalization, Equation (4), so we must estimate the rates of loss to space and organic production.

The rate of loss to space $k_{loss}$ can be estimated assuming an XUV-dominated escape as

$$k_{loss} = \frac{\epsilon_{XUV} \, L_{XUV} \, R_{terr}^3}{M_{H_2} \, a_{temp}^2 \, G \, M_{terr}} \tag{32}$$

Note that this rate depends linearly on total atmospheric hydrogen, as the energy-limited escape flux does not depend on atmospheric mass. The dependence of the X-ray luminosity for an early star $L_{XUV}$ was found in [10], and the efficiency factor $\epsilon_{XUV}$ is likely not to depend on constants to any meaningful extent, so the dependence of $k_{loss}$ is completely determined.

If we take the size of the impactor to be the isolation mass, this defines a timescale, which is normalized to be 10 Myr [17]:

$$t_{reduced} = 2.2 \times 10^{10} \, \frac{\kappa^{3/2} \, \lambda^{121/40} \, m_p^{19/4} \, M_{pl}^{1/4}}{\alpha^{27/2} \, m_e^6} \tag{33}$$

The dependence of $k_{prebio}$ is more difficult to estimate, as it stems from the production (and destruction) of multiple preorganic compounds through a complex chemical reaction network in the atmosphere, as well as the rainout rate of produced molecules. In practice, we may estimate the production rate by examining the production of formaldehyde ($H_2CO$), which is the dominant organic molecule produced [73] and is expected to play a crucial role in the formation of sugars [74]. The reaction pathway for this molecule may be summarized as $CO_2 + 2H_2 \rightarrow H_2CO + H_2O$ [75], though each individual molecular interaction depends only on the interaction with a single hydrogen molecule, or ion thereof. Therefore, we expect the overall rate to scale as the limiting step within this process, which will scale as $k_{slowest} \propto n_{H_2}/(T^3 \, \tau_{int})$, where the interaction time[3] is $\tau_{int} \sim L/v \sim n^{-1/3}/\sqrt{T/m}$. This can then be related to the total atmospheric hydrogen through $n_{H_2} = g \, M_{H_2}/(4\pi R_{terr}^2)$ to be

$$k_{prebio} \sim \frac{G^{4/3} \, M_{atm}^{4/3} \, M_{terr}^{4/3}}{(4\pi)^{4/3} \, R_{terr}^{16/3} \, m_p^{1/2} \, T_{temp}^{23/6}} \tag{34}$$

This scales super-linearly with $M_{atm}$ on account of the fact that a larger atmosphere will be denser, decreasing the typical time between interactions. In the limit $k_{prebio} \ll k_{loss}$, we then have $p_{life} \propto \Delta S^{4/3}$. In this limit, we have the usable disequilibrium

$$\Delta S_{moneta} = 6.1 \times 10^{-18} \frac{\kappa^{4/3} \lambda^{619/60}}{\alpha^{34} \beta^{55/4} \gamma^{3/2}} \tag{35}$$

## 5. Panspermia

We now turn to the final scenario for the origin of life which we consider, panspermia. According to this hypothesis, life may not have originated on Earth but instead could have been delivered from elsewhere (see [76] for a recent review). As such, it is not actually a theory of the origin of life per se, but only a theory for the origin of life on Earth. Though the other scenarios could be included in our calculations by their effect on the probability of the emergence of life $p_{life}$, this scenario differs in that it fully modifies the Drake equation to take into account transfer of life between systems.

An argument for the panspermia hypothesis comes from the fact that early environments on other planets, such as Mars, may have been more clement and conducive to the origin of life that early Earth [77]. This, coupled with the observation that a substantial amount of material is ejected from a planet during impacts, leads to a plausibility argument that life could have been exchanged between worlds [78]. However, at the moment, it is highly uncertain whether life could survive the harsh conditions during ejection, transit through space, and reentry [79].

There are essentially two variants of the panspermia hypothesis: interplanetary panspermia and interstellar panspermia. We will consider each in turn. To begin, however, we consider the problem in the abstract to determine the effects panspermia has on the expected number of planets harboring life.

We consider a system of $n$ planets, each of which have a probability for the emergence of life $p_{life}$. If life does emerge on a planet, the probability that it is transferred to a lifeless world is denoted as $p_x$. In this setup, all planets are treated as having identical values for these probabilities. We wish to consider the expected number of planets harboring life at the end of the system's evolution. In the limit $p_x \to 0$, panspermia is not operational, and we have $\langle n_{life} \rangle \to n\, p_{life}$. In the opposite limit $p_x \to 1$, where if life does emerge on a planet, it is guaranteed to be transferred to every other planet in the system, we have $\langle n_{life} \rangle \to n(1 - (1 - p_{life})^n)$. This indicates that as long as life arises on any planet in the system, it will be present on all $n$ of them. Notice that, in this expression, in the limit where life is very rare, $p_{life} \to 0$, the expected number tends to $\langle n_{life} \rangle \to n^2 p_{life}$, so that the dependence on the number of planets in the system is enhanced. Also note that, though we restrict our consideration to the same class of objects, if the sites where life could possibly originate are different than the sites where complex life can take hold, the two factors could be different, so that $\langle n_{life} \rangle \to n_{originate} n_{thrive} p_{life}$. This would be especially relevant if the origination sites were much more numerous than temperate, terrestrial planets, such as the possibility that life may have arisen on smaller icy bodies in the outer system [80].

The expression for the average number of planets with life in intermediate regimes is a complicated polynomial, $\langle n_{life} \rangle = n \sum_{k=1}^{n} c_{k,n}(p_x) p_{life}^k$. The coefficients $c_{k,n}$ are polynomials that depend on $k$ and $n$ and are determined by the combinatorics of independent origin versus transfer in a multi-system setup[4]. At present, the authors can find no general expression valid for all values of $n$. This, coupled with the fact that, in general, the expected value of $n$ in our setup will actually be a non-integer, leads us to suggest the following approximate formula that may be used, containing the relevant asymptotics:

$$\langle n_{life} \rangle \approx n \left[ p_{life} + p_x \left( 1 - p_{life} - (1 - p_{life})^n \right) \right] \tag{36}$$

Again, in the limit where $p_{life} \to 0$, this tends toward $\langle n_{life} \rangle \to n((1 - p_x) + p_x n) p_{life}$.

This can be incorporated into our analysis by modifying our expressions for habitability $\mathbb{H}$ (Equation (1)) in the following ways: for interplanetary panspermia, habitability is augmented (in the limit $p_{\text{life}} \ll 1$, as discussed above) to be

$$\mathbb{H} = N_\star \left( (1 - p_x) + p_x\, n_e \right) n_e\, p_{\text{life}}\, p_{\text{int}}\, N_{\text{int}} \tag{37}$$

In general, each quantity in this expression (and those that follow throughout this section) will depend on the fundamental constants, but this dependence is left implicit in our formulas. This quantity is averaged over stellar masses (and, where relevant, other variables, such as the birth time of the system) to determine the overall habitability of a universe. In [5], we found that the expected number of rocky planets per star is $n_e = 0.0061\alpha^{4/3}\beta^{13/16}/(\kappa^{1/2}\lambda^{5/6})$.

For interstellar panspermia, more care is needed in delineating the system where material exchange can be expected. For this, we must decompose the number of stars in the universe into the number of galaxies of a given mass,

$$N_\star = N_{\text{gal}} \int dM_{\text{gal}}\, p(M_{\text{gal}})\, N_{\text{stars/gal}}(M_{\text{gal}}) \tag{38}$$

If we take the distribution of galaxy masses to be of the form $c(M_{\text{gal}}) = \text{erf}((M_{\text{gal}}/(\sqrt{\pi}\langle M_{\text{gal}}\rangle))^{1/3})$, as found in [81], we find the modified count of the number of observers can be expressed as

$$\mathbb{H} = N_\star \left( (1 - p_x) + \frac{15\pi}{8} p_x \langle N_{\text{stars/gal}}\rangle \right) n_e\, p_{\text{life}}\, p_{\text{int}}\, N_{\text{int}} \tag{39}$$

In [82], the typical number of stars per galaxy was found to be $\langle N_{\text{stars/gal}}\rangle = 4.35\alpha^5/(\beta^{1/2}\gamma)$. The precise cofactor for the second term in this expression is specific to the galaxy mass distribution we have used but will play no role in the limiting case we consider, where the second term is much larger than the first.

Above, we considered exchange to be possible between any two stars of a given galaxy, though panspermia may be much more likely to occur in a star's higher density birth cluster if life arose before its dissipation [83]. If, instead, we wish to consider stellar clusters to be our systems of consideration, a similar decomposition can be made. The key difference is that the distribution of stars per cluster is better described by a power law, $p(N_{\text{stars/cluster}}) \propto 1/N_{\text{stars/cluster}}^2$ [84]. As such, the expected value of the square of the system size is not set by the average value but rather the largest clusters, the size of which are dictated by their host galaxies. Therefore, even in this scenario, the modifications to the expected number of observers will be of the same form as Equation (39) above.

To complete our analysis, we also analyze how $p_x$ may change with constants. Generically, we expect the exchange probability to depend on the number of rocks exchanged between planets $N_x$ and probability that a single exchange seeds life $p_{\text{seed}}$ as $p_x = 1 - (1 - p_{\text{seed}})^{N_x} \to p_{\text{seed}} N_x$. This latter approximation, valid in the limit where $p_x$ is small, will suffice for our analysis. Here, we treat $p_{\text{seed}}$ as some unknown biological quantity that plausibly does not depend much on the physical constants and elaborate on how the factor $N_x$ depends on constants, mostly following [83].

The number of suitable rocks ejected from planetary bodies during solar system evolution is $N_{\text{ejected}} \approx (m_{max}/m_{min})^q$, where $q \approx 3/4$ is a coefficient determining the size distribution of impact ejecta [85,86]. The maximum rock mass is set by the isolation mass $M_{\text{max}} \sim M_{\text{iso}}$. The minimum viable mass is set by the amount needed to shield biological material present in the interior of the rock from UV radiation, which determines the minimum radius by $\tau_{UV} \sim 2\pi^5/3a_{\text{Bohr}}^6/\lambda_{UV}^4 \rho/m_p L_{\text{min}} \sim 1$. This gives $L_{\text{min}} \sim 1/(a_{\text{Bohr}}^3 \text{Ry}^4)$, or $m_{\text{min}} = 136.3 m_p/\alpha^{12}$.

For interplanetary panspermia, the number of suitable rocks exchanged between planets is then $N_x \approx N_{\text{ejected}} f_{\text{hit}}$, where $f_{\text{hit}}$ is the ratio of planetary to stellar cross-sections discussed above. If we normalize this to $N_{\text{ejected}} = 10^{16}$ [85] and $f_{\text{hit}} = 10^{-7}$ [59], this yields

$$p_x^{\text{interplanetary}} = 8.9 \, p_{\text{seed}} \frac{\kappa^{9/8} \alpha^{75/8}}{\lambda^{33/160} \beta^{7/4} \gamma^{35/16}} \tag{40}$$

For interstellar panspermia, this must be multiplied by an additional factor $\tau_{\text{cluster}} = \langle \sigma v \rangle t_{\text{cluster}} n_{\text{cluster}}$ representing the optical depth for exchange of material between two stars. Here, we have specified that the exchange is most likely to occur while stars are in their birth cluster. As discussed in [83], this presupposes that life has enough chance to arise during this phase. This is a strong assumption, since cluster lifetimes are 10–100 Myr. This is shorter than the development timescale suggested even by the earliest evidence of life on Earth, at 4.2–3.8 Ga [87,88], but it has been argued that this timescale could be sufficient for the development of living systems [89].

The velocity-averaged cross-section can be estimated as $\langle \sigma v \rangle \sim \pi a_{\text{ice}}^2 v_{\text{cluster}}$, where we have used that locations of the outer giant planets are expected to be set by the ice line $a_{\text{ice}}$ [90]. The cluster dispersion speed is set by the molecular cooling temperature, $v_{\text{cluster}} \sim \sqrt{T_{\text{mol}}/m_p}$ [91]. This also dictates the cluster lifetime as set by the free fall time $t_{\text{cluster}} \sim 1/\sqrt{G \rho_{\text{cluster}}}$ and the stellar density $n_{\text{cluster}} = \rho_{\text{cluster}}/M_\star$ as given by $\rho_{\text{cluster}} \sim v_{\text{cluster}}^6/(G_N^3 N_{\text{cluster}}^2 M_\star^2)$, where $N_{\text{cluster}}$ is the number of stars in the cluster. This gives

$$\tau_{\text{cluster}} \sim \pi \, a_{\text{ice}}^2 \frac{v_{\text{cluster}}^4}{G_N^2 N_{\text{cluster}} M_\star^2} \sim \frac{\alpha^{2/3} \beta^{1/2}}{N_{\text{cluster}}} \tag{41}$$

A full calculation of the exchange probability $p_x$ would average this optical depth over cluster sizes, but for our purposes, we normalize $\tau_{\text{cluster}} = 10^{-4}$, from [83]. We then have

$$p_x^{\text{interstellar}} = 4.4 \times 10^{-6} \, p_{\text{seed}} \frac{\kappa^{9/8} \alpha^{361/24}}{\lambda^{33/160} \beta^{7/4} \gamma^{51/16}} \tag{42}$$

With these modifications to the Drake equation, we may treat $p_x \, n_e$ and $p_x \, N_{\text{stars/gal}}$ as effective terms that encapsulate the efficacy of each scenario in much the same way that $p_{\text{life}}$ did for the other origin of life scenarios. Additionally, it would be possible to combine the panspermia scenario with the other origin of life scenarios, but here, we treat them as separate, and only consider $p_{\text{life}} = \text{const}$ in the panspermia scenario.

## 6. Synthesis

Having derived expressions for the disequilibrium generated in each origin of life scenario, we now are able to incorporate each into our multiverse probability calculations. This allows us to determine whether to expect the probability of life's emergence to depend on the total amount of disequilibrium or not and which scenarios are favored within the multiverse framework. To begin, we assemble the expressions we found for total disequilibrium produced, which under our ansatz correspondingly yield the probability of life's emergence, in Table 1.

One feature of note that can be gleaned from this table is the commonality in $\gamma$ dependence; most of these factors depend on $\gamma$ as $1/\gamma^q$, where $q$ is usually between 3 and 4. This is at first surprising, given that these sources of disequilibrium are all so physically disparate. However, this happenstance can be understood as follows: there are many sources of disequilibrium in the universe. This is not an exhaustive list, merely the ones which have been proposed as potentially relevant for the origin of life because they are capable of supplying a large amount of disequilibrium. Therefore, these candidates represent the top of a ranked list of all disequilibrium sources (though Table 1 places scenarios in order of presentation within the text and not magnitude). However, given that all disequilibrium sources can be estimated as an algebraic function of the physical

constants, and given the fact that the constant $\gamma$ is by far the smallest of the physical constants, then naturally the ordering of disequilibrium sources would sort them by their dependence on this constant. Indeed, obtaining a factor on the order of $10^{\approx 45}$ from a combination of these small constants necessitates this $\gamma$ dependence. One consequence of this observation is that if the origin of life plays a determining factor in our placement within the multiverse, then, no matter the scenario, this factor exerts a fairly strong pressure toward universes with smaller $\gamma$, i.e., weaker gravity.

**Table 1.** Disequilibrium production for the different origin of life scenarios considered. For scenarios where we expect deviations from the ansatz $p_{\text{life}} \propto \Delta S$ (the Moneta and panspermia scenarios, see main text), an effective $\Delta S$ is presented as the factor which modifies the usual probability calculations.

| Scenario | $\Delta S$ | Source of Disequilibrium Production |
|---|---|---|
| Lightning | $1.1 \times 10^{-5}\, \tilde{\epsilon}_{\text{lightning}} \lambda^{-5/2}\, \alpha^7\, \beta^{3/2} \gamma^{-4}$ | Lightning flashes |
| SEP | $0.053\, f_{\text{open}}^{-1}\, \lambda^{-43/10}\, \alpha^{12}\, \beta^{17/12}\, \gamma^{-11/3}$ | Solar energetic particles |
| XUV | $3.9 \times 10^{-8}\, f_{\text{open}}^{-1}\, \lambda^{-23/20}\, \alpha^{17/2}\, \beta^{3/2}\, \gamma^{-4}$ | High-energy solar photons |
| Hydrothermal vents | $66.5\, f_{\text{vol}} \lambda^{-5/2}\, \alpha^{9/2}\, \gamma^{-3}$ | Hydrothermal material from oceanic vents |
| IDP | $1.2 \times 10^{-5}\, \alpha^6\, \beta^{1/2}\, \lambda^{-31/20}\, \gamma^{-7/2}\, \theta_{\text{chem}}$ | Organic material from IDPs |
| Comets | $5.4 \times 10^{-7}\, \alpha^9\, \beta\, \lambda^{-23/10}\, \gamma^{-4}$ | Material created during shock synthesis |
| Asteroids | $0.94\, \kappa\, \lambda^{21/10}\, \alpha^{-4}\, \beta^{-4/3}\, \gamma^{-8/3}$ | Material created during shock synthesis |
| Moneta | $6.1 \times 10^{-18}\, \kappa^{4/3}\, \lambda^{619/60}\, \alpha^{-34}\, \beta^{-55/4}\, \gamma^{-3/2}$ | Large impact triggered reducing atmosphere |
| Interplanetary panspermia | $\kappa^{5/8}\, \lambda^{-499/480}\, \alpha^{257/24}\, \beta^{-15/16}\, \gamma^{-35/16}$ | Transfer of life between planets |
| Interstellar panspermia | $\kappa^{9/8}\, \lambda^{-33/160}\, \alpha^{361/24}\, \beta^{-7/4}\, \gamma^{-51/16}$ | Transfer of life between star systems |

We now incorporate each of these into our probability calculations. The main challenge in encapsulating the effect each disequilibrium source may have on our probabilities is that there are a variety of other hypothetical habitability criteria, each with uncertain status. While including these origin of life factors decreases the likelihood of some combinations of habitability criteria, it also raises the likelihood of other combinations. To adequately determine the effect each source of disequilibrium may have, we therefore combine them with a moderately large set of previously studied hypotheses. These are as follows:

- **photo** and **yellow**: complex life requires photosynthetically active starlight, with optimistic and pessimistically defined ranges, respectively [4].
- **TL**: complex life requires the planet to be tidally unlocked [4].
- **bio**: complex life requires the star to last for a biological timescale [4].
- **terr**: complex life requires a terrestrial planet, $v_{\text{esc}}^2 \sim T/m_p$ [5].
- **temp**: complex life requires a temperate planet, $T \sim E_{\text{mol}}$ [5].
- **plates**: complex life requires radiogenic plate tectonics [6].
- **time**: the emergence of complex life is proportional to the stellar lifetime [6].
- **area**: the emergence of complex life is proportional to the planet area [6].
- **S**: the emergence of complex life is proportional to the incident radiation flux [6].
- **C/O**, **Mg/Si**: complex life requires a specific C/O or Mg/Si ratio [8].
- **N**: complex life requires sufficient nitrogen [8].
- **obliquity**: complex life requires stable obliquity [9].

This is not an exhaustive list of the habitability criteria we have investigated previously but represents the usually most impactful. Even combining this restricted list with our ten origin of life scenarios (plus the additional null hypothesis, where the probability of the emergence of life does not depend on the physical constants) in full generality leads to 101,376 combinations. To alleviate the computational burden of this exhaustive combinatorial profusion, we restrict our combinations to a depth of four. That is, for any given run, we assume that at most four from the list of habitability criteria/origin of life scenarios are true, while the others are false. Even with this restricted search, this leads

to 5754 combinations. Note that here, we treat each of these habitability conditions as logically independent of the origin of life scenarios we consider. It may be that certain combinations are incompatible, for instance, that hydrothermal vents either require or are greatly enhanced by the presence of plate tectonics. Our strategy has been to report all possible combinations as they occur, but if some can be argued to be inconsistent, these may be disregarded.

We remind the reader that we explicitly assume temperate, terrestrial planets for many of the calculations we perform in this text, and so specifically exclude other locales where life may arise, such as giant planets or moons. This assumption either amounts to assuming life may only arise on Earth-like planets or restricting our "reference class" in the sense of [92] to observers that arise on these planets. In either interpretation, this is seemingly only compatible with assuming both the **terr** and **temp** conditions. The effect these have is to weight universes according to the fraction of planets within the terrestrial and temperate ranges, respectively. For the terrestrial condition, this is the fraction of planets within 0.3 to 4 Earth masses (defined using the stability of light and heavy atmospheric constituents). For the temperate condition, this weights universes by the ratio of the habitable zone to the interplanetary spacing [5]. Here, we include these habitability conditions as optional throughout this section, but a fully consistent analysis would need to determine formulas for the probability of the emergence of life on a broader range of planets, which is left for future work.

For each combination we consider, we compute the probability of eight observations: the five physical constants mentioned in the introduction ($\alpha$, $\beta$, $\gamma$, $\delta_u$ and $\delta_d$), as well as the probability of orbiting a star as massive as our sun, the probability of measuring our value of the Hoyle resonance energy, and the probability of observing our value of the organic-to-rock ratio.

The probability of orbiting a star at least as massive as ours is an important way to assess a habitability condition and does not rely on the multiverse at all. It is defined through $P(\lambda_\odot) = \int_{\lambda_\odot}^\infty p_{\text{IMF}}(\lambda)\mathbb{H}(\lambda)$, with all fundamental constants fixed to their observed values. It has proven useful toward penalizing theories that highly favor low-mass stars [6,93].

The other two observables we consider are not fundamental constants but are very important macroscopic features of our universe and have historically played a pivotal role in the idea of the multiverse. The first is the probability of observing such a small value of the Hoyle resonance energy $E_R$, defined as the energy difference between a particular excited state of carbon and the ground state energy of three helium nuclei. As first pointed out in [94], this energy dictates the process of nuclear burning in stars, and its small positive observed value is directly responsible for the fact that carbon is abundant in our universe. Computing the probability of observing such a small value in the multiverse framework is one way of encapsulating the selection effect that would give rise to this observation for habitability criteria that require carbon (for more details see [8]). This is equivalent to the probability of observing a carbon-to-oxygen ratio at least as large as what we observe.

The last supplemental observable we consider is the probability of observing an organic-to-rock ratio, defined as the ratio of element abundances $R_{o/r} = (C + O)/(Mg + Si)$, to be at least as large as ours. As described in [8], based off the results in [53], to a first approximation, this quantity is also only dependent on $E_R$, but this peaks near our observed value. It serves as another useful diagnostic for why our universe is unusually rich in organic matter and favors habitability criteria that account for this.

These eight observables serve to indicate how well a given combination of habitability conditions and origin of life scenario account for our observations, both in the multiverse and within our universe. As a summary statistic, we consider the product of all these values as the (naive) Bayes factor $\mathcal{B}$ as a function of habitability condition $\mathbb{H}$:

$$\mathcal{B}(\mathbb{H}) = \mathbb{P}(\alpha|\mathbb{H})\,\mathbb{P}(\beta|\mathbb{H})\,\mathbb{P}(\gamma|\mathbb{H})\,\mathbb{P}(\delta_u|\mathbb{H})\,\mathbb{P}(\delta_d|\mathbb{H})\,\mathbb{P}(\lambda|\mathbb{H})\,\mathbb{P}(E_R|\mathbb{H})\,\mathbb{P}(R_{o/r}|\mathbb{H}) \quad (43)$$

As a baseline, the combination of habitability criteria that do not account for the origin of life factor with the highest Bayes factor is **TL + bio + area + C/O**, with a value of

$5.91 \times 10^{-5}$. Though this may seem a small number, bear in mind that it is the product of 8 separate probabilities, which have an average value of 0.36. In the following, the Bayes factors for all scenario combinations are reported relative to this baseline. Again, we stress that these are computed explicitly, assuming the principle of mediocrity as a starting point, where the probability of an observation is proportional to the number of observers that make such an observation. It is important to note that alternatives to this assumption do exist [95,96].

The distribution of Bayes factors is plotted in Figure 1. From here, it can be seen that many combinations of habitability criteria lead to extremely low likelihoods of our observations within the multiverse framework and so are incompatible with the multiverse. We also observe that including the origin of life scenarios does not appreciably alter the distribution of Bayes factors. Thus, on the whole, the idea that the emergence of life acts as a bottleneck, favoring universes which are most prolific in their production of life, is compatible with, but not required in, the multiverse framework.

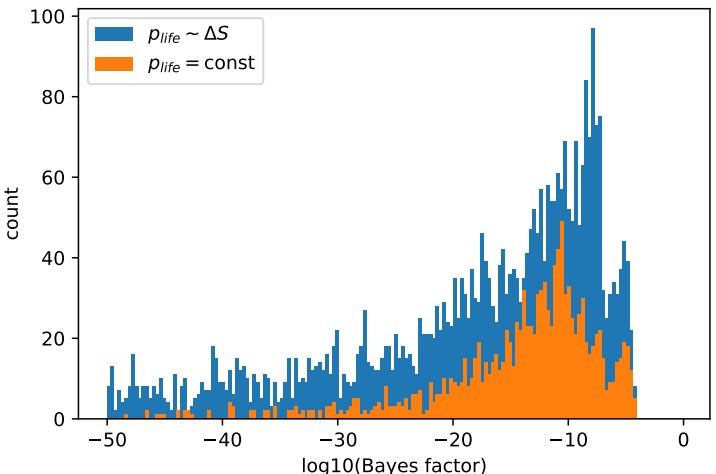

**Figure 1.** Histogram of Bayes factors, representing the probability of our observations within the multiverse framework for different habitability criteria combinations. These are computed by compounding the probabilities of the eight observations described in the text. While some combinations lead to reasonable probabilities (the largest having a total value of $5.91 \times 10^{-5}$, corresponding to an average probability of 0.36 per observation, out of a maximum 0.5), others are extraordinarily low.

While the overall hypothesis that the emergence of life depends linearly on disequilibrium produced is neither favored nor disfavored by the multiverse, the various sources of disequilibrium have varying levels of support within the multiverse framework. In Table 2, we display the best combination of habitability hypotheses for each disequilibrium source considered in the text, along with the associated Bayes factor relative to the null hypothesis. Additionally, in Figure 2, we show a pair plot combining each origin of life scenario with each of the other habitability criteria. The values in this plot are the maximum Bayes factors for all criteria combinations containing both named criteria. The values have been normalized to the maximum Bayes factor for the null hypothesis. From this, it can be seen that several of the disequilibrium sources are disfavored. The two most disfavored theories are the interplanetary and interstellar panspermia theories, as evidenced by the fact that all entries in the last two rows of Figure 2 are 0.05 or less. This indicates that if either of these scenarios are true, then our presence in this universe is quite unlikely. This is due to the fact that for other values of the physical constants, panspermia is much more likely, leading to an overwhelmingly larger number of life-bearing systems. Several other theories, such as the SEP and hydrothermal vent sources, are also disfavored, but to a lesser extent, and no strong conclusions should be drawn for these. The asteroids source actually enhances the likelihood of being in our universe relative to the baseline case. This provides several predictions for which of the origin of life scenarios are true given that the

multiverse hypothesis is true. If subsequent observations indicate that the correct origin of life scenario does not line up with the multiverse predictions, this will count as negative evidence for the existence of a multiverse.

**Table 2.** Best combinations of habitability criteria for each origin of life scenario, along with their associated Bayes factor, relative to the null hypothesis.

| Criteria | Best Combination | $\mathcal{B}$ |
|---|---|---|
| - | **TL bio area C/O** | 1.0 |
| Lightning | **photo TL C/O lightning** | 0.89 |
| SEP | **C/O SEP** | 0.002 |
| XUV | **TL C/O obliquity XUV** | 0.057 |
| Hydrothermal vents | **photo TL C/O vents** | 0.21 |
| IDP | **TL C/O IDP** | 0.53 |
| Comets | **photo TL C/O comets** | 0.36 |
| Asteroids | **TL temp C/O asteroids** | 1.18 |
| Moneta | **yellow C/O terr Moneta** | 0.004 |
| Interplanetary panspermia | **yellow plates C/O plan. pans.** | 0.014 |
| Interstellar panspermia | **yellow plates C/O stel. pans.** | 0.045 |

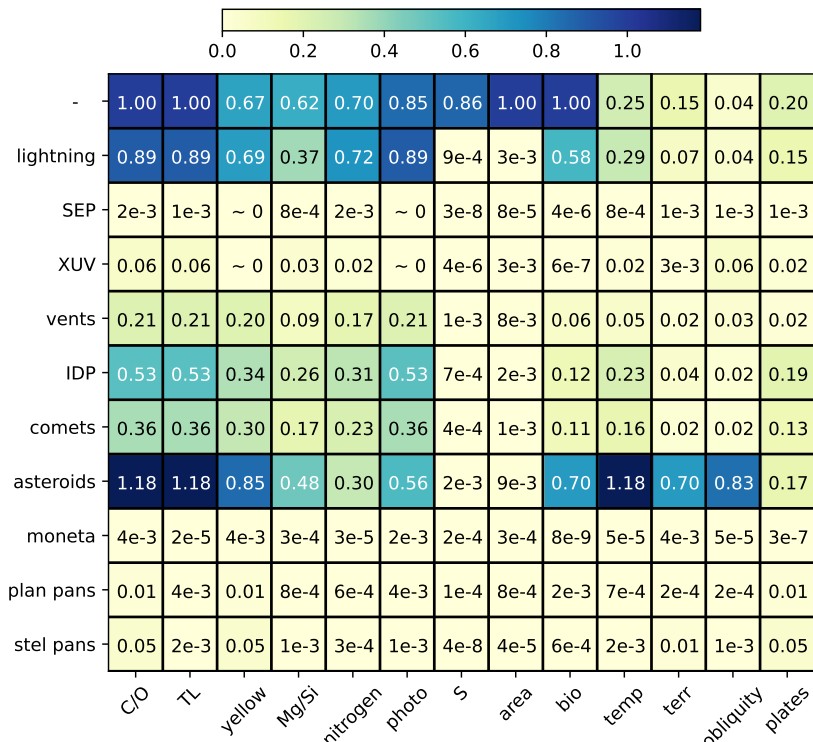

**Figure 2.** Bayes factors for different combinations of habitability criteria. Each entry in this matrix represents the best combination of criteria containing the origin of life scenario and habitability condition specified on their respective axes. Here we use the notation ae-x= $a \times 10^{-x}$.

One additional feature that emerges from this analysis is the interplay between $p_{\text{life}}$ and $p_{\text{int}}$, the fraction of biospheres that develop intelligence. Our previous works had favored $p_{\text{int}} \propto \Delta S_{\text{rad}}$, the total incident radiation on a planet (the column marked $S$ in Figure 2). The rationale for this is that this quantity controls the overall size of the biosphere [6] (again, discounting any potentially damaging reactions, an excess of radiation may incur), and so plausibly is related to the probability of the emergence of intelligence on a planet. However, this hypothesis is incompatible with the origin of life being proportional to the disequilibrium in the multiverse framework. When both these factors are included in the calculation, the maximum likelihood scenario, **TL S Mg/Si area**, is disfavored by a factor of

0.0023 relative to the baseline case. This is because the two factors both favor small $\gamma$. When combined, the pressure toward lower values of $\gamma$ is too strong to account for our presence in this universe. Additionally, for this very reason, origin of life scenarios that depend on disequilibrium as $p_{\text{life}} \propto \Delta S^n$, with $n > 1$, are disfavored in the multiverse framework.

We also give some indication of the relative contribution of the eight observables we consider to these results by noting the number of times each variable has the lowest probability. Of the 5754 habitability combinations we consider, these are: $\alpha$: 326, $\beta$: 541, $\gamma$: 767, $\delta_u$: 1, $\delta_d$: 11, $\lambda_\odot$: 1508, $E_R$: 3, $R_{o/r}$: 2597. This indicates that the organic-to-rock ratio is the most constraining of the observables we consider, and that the probability of orbiting a sun-like star is the second most, even though it is independent of the multiverse. Of the five fundamental constants we consider, the strength of gravity is the most constraining.

It is worth pausing here to discuss the implications of this. We have found that within the multiverse framework, either $p_{\text{life}}$ is proportional to $\Delta S$, or $p_{\text{int}}$ is proportional to $\Delta S$, or neither are, but not both. In the introduction, we argued that if life were sensitive to the amount of disequilibrium produced, it would indicate that the fraction of planets possessing life is closer to 0 than 1. Under the simplest interpretation, the same may be said of the fraction of biospheres that evolve intelligence. With this, we would conclude that both fractions cannot be sensitive within the multiverse framework, and so at least one must be insensitive to the amount of disequilibrium produced and therefore close to 1. However, this conclusion is not inescapable, and to illustrate this, we briefly entertain an alternative scenario. Recall the rationale for intelligence to depend linearly on $\Delta S$ was that the total biosphere size scales linearly with this quantity, assuming that primary production maximizes photosynthetic intake. However, the probability of intelligence emerging may not scale linearly with biosphere size. Indeed, we may instead expect that the probability of intelligence emerging is proportional to the number of multicellular species. Extrapolating from observed species-abundance distribution models indicates that the number of species only depends logarithmically on biosphere size [97]. This alternative account of the emergence of intelligence is consistent with both the multiverse and the expectation that both $p_{\text{life}}$ and $p_{\text{int}}$ are much less than 1.

One last point to note, which can be observed in Figure 2, is the fact that certain habitability criteria which were not viable with the null hypothesis become viable when taking certain origin of life scenarios. This effect is most notable for the obliquity criterion, which is highly disfavored when not including $p_{\text{life}}$ ($\mathcal{B} = 0.04$), but becomes consistent with the asteroids scenario ($\mathcal{B} = 0.83$). This can be seen to a lesser extent for the plate tectonics and temperate zone criteria as well, where it can be noted that the XUV disequilibrium source is most viable when plate tectonics is taken to be necessary for complex life. This highlights how the interplay between different habitability criteria can alter the probability of our observations in nontrivial ways. As such, we are usually prevented from making blanket statements on whether certain habitability criteria are favored or disfavored. We are instead forced to make more qualified statements that certain combinations of habitability criteria are (in)compatible with the multiverse hypothesis to a quantified degree of statistical certainty. Though this additional nuance may prevent us from forming succinct slogans for our predictions, it does not alter the fact that concrete, testable predictions can be generated by the multiverse framework.

**Author Contributions:** Conceptualization, all authors; Methodology, M.S.; Formal Analysis, M.S.; Validation, V.A., L.B., G.F.L. and I.P.-R.; Writing—Original Draft Preparation, M.S.; Writing—Review & Editing, V.A., L.B., G.F.L. and I.P.-R. All authors have read and agreed to the published version of the manuscript.

**Funding:** This research received no external funding.

**Data Availability Statement:** All code to generate data and analysis is located at https://github.com/mccsandora/Multiverse-Habitability-Handler., accessed on 10 Ooctober 2022.

**Conflicts of Interest:** The authors declare no conflict of interest.

## Appendix A. Sensitivity Analysis

Here, we determine how sensitive our calculations are to the choice of two inputs: the choice of prior for the fundamental constants, and the solar energetic particle energy distribution parameter $k_{SEP}$. For the SEP analysis, we range over the full habitability conditions we consider in the full text, except for the fact that we restrict ourselves to the **SEP** origin of life scenario, resulting in 444 total combinations. In our analysis of the sensitivity to the prior, for expediency, we restrict ourselves to a smaller subset of the more impactful habitability conditions to vary: the **photo**, **yellow**, **TL**, **time**, **area**, **S**, **C/O**, **Mg/Si**, and **N** conditions. When simultaneously varied over the ten origin of life scenarios, this amounts to 1339 different combinations. We compare the total Bayes factor originally calculated to the Bayes factor of the altered variable.

The prior for the fundamental constants we use throughout the main text is log-uniform for each mass ratio we consider, and uniform for the force strength $\alpha$, yielding $p_{prior} \propto 1/(\beta \gamma \delta_u \delta_d)$. This has a separate motivation for each factor. First, the scale of the proton mass can be obtained from the strong force couplings through the process of dimensional transmutation (see [98] for a discussion in the multiverse context), where it can be seen that the dependence is exponential in the couplings, leading to a log-uniform prior (up to potential logarithmic corrections, depending on the prior taken for the coupling). For the other three mass ratios, a log-uniform distribution was argued in [99] to be observed of standard model particle masses, though the statistics are limited and favored on theoretical grounds as the distribution preserved under renormalization group flow. Thus while we believe these to be reasonably well founded, here, we explore the impact of instead choosing uniform priors for each of the mass ratios. In the absence of information, we have taken $p_{prior}(\alpha)$ to be uniform. However, we may also expect the underlying theory to have a uniform distribution of charges, rather than force strength, which would imply $p_{prior}(\alpha) \propto 1/\sqrt{\alpha}$, and so we explore this possibility here.

In Figure A1, we vary these assumptions one at a time to gain an indication for how important each is. The leftmost plot compares the Bayes factors computed for each prior assumption, where each point is a different combination of habitability criteria. The right plot shows the ratio of the new to original Bayes factors displayed against the original Bayes factor. Though the scatter in these plots dictates that the resultant Bayes factor may vary by several orders of magnitude depending on assumptions we make about the prior, this scatter is seen to decrease for larger original Bayes factors, where the difference would be most important. Additionally, this scatter, while it would certainly affect some of our conclusions, is paltry in comparison to the differences obtained when the measure of cosmological parameters is varied (see, e.g., [100]).

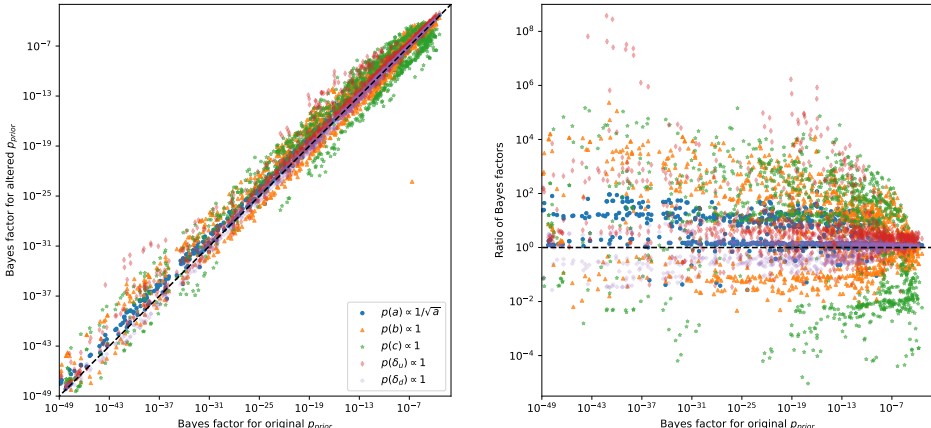

**Figure A1.** Comparison of Bayes factor using standard prior with Bayes factors using altered priors. Each dot is a combination of habitability criteria. The left plot compares the two Bayes factors directly and indicates a strong correlation with some scatter. The right plot divides the new Bayes factor by the original to better indicate the residual scatter.

We also note that the Bayes factor depends on the prior for some variables more than others. To quantify this, we report the average absolute multiplicative change induced, defined as $\log a = \langle|\log(\mathcal{B}_{\text{altered}}/\mathcal{B}_{\text{original}})|\rangle$. Then, we have $\alpha: 2.78$, $\beta: 22.8$, $\gamma: 91.3$, $\delta_u: 8.12$, $\delta_d: 1.14$, which indicates that, for instance, on average, the change in Bayes factor for the altered $\alpha$ prior is a factor of 2.78 times smaller or larger than the original. This is seen to depend on the prior for $\gamma$ most sensitively, which is expected, since the range of parameter space for the strength of gravity extends several orders magnitude upward for many habitability conditions [4,82]. However, we note that the original arguments for this quantity being log-uniform are most standard.

In Figure A2, we repeat the above analysis for different choices of the SEP scaling coefficient $k_{\text{SEP}}$. To illustrate the dependence on this quantity, we choose two additional values to the original choice of 1, which are $1/2$ and $2$. The average multiplicative changes here are $k_{\text{SEP}} = 2:108.7$, $k_{\text{SEP}} = 1/2:7.67$. Here, though there is some shift in Bayes factors, with a systematic preference for smaller $k_{\text{SEP}}$, this shift is not large enough to alter our conclusions that this origin of life scenario is disfavored within a multiverse framework.

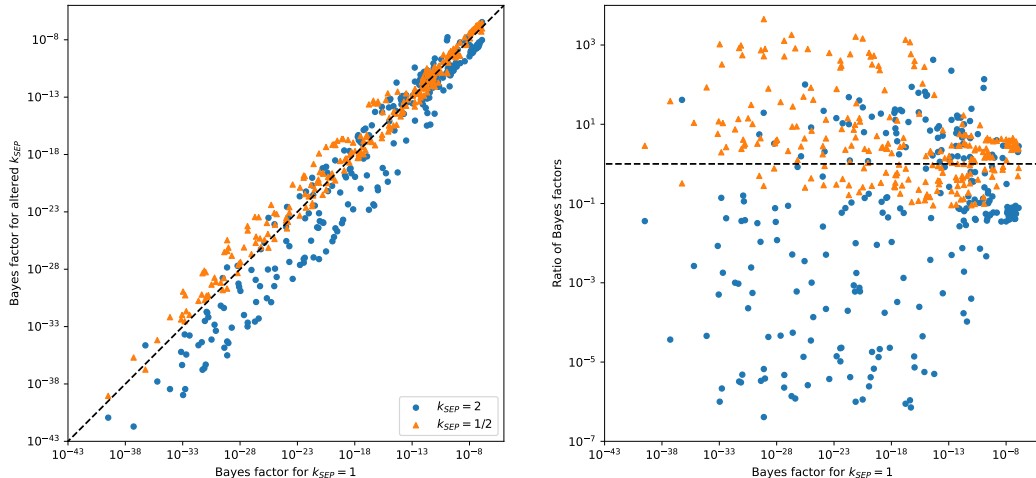

**Figure A2.** Comparison of Bayes factor using $k_{\text{SEP}} = 1$ with Bayes factors using altered $k_{\text{SEP}}$. Each dot is a combination of habitability criteria. The left plot compares the two Bayes factors directly and indicates a strong correlation with some scatter. The right plot divides the new Bayes factor by the original to better indicate the residual scatter.

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
