# Peer review of "Multiverse Predictions for Habitability: Origin of Life Scenarios"

_universe, doi:10.3390/universe9010042_

Round 1
Reviewer 1 Report
The paper is a continuation of a series of papers focused on the habitability
predictions from a multiverse perspective. The issues previously discussed comprise the number of habitable stars, potentially habitable planets and fractions of them which develop life and inteligence. The present paper discusses scenarios for the origin of life. The underlying principle is that if life is rare and sensitive to specific conditions, then we should expect living in a universe which is better than average at creating these conditions. This is a form of anthropic principle reasoning. In order to make it operative, the authors propose to express the probability of life as a function of chemical disequilibrium. They successfully propose a framework allowing to express probability of life in different scenarios in terms of fundamental
constants like: the fine structure constant the ratio of electron to proton mass, the strength of gravity, and the ratios of the up and down quark masses to the proton mass. This is a very nice idea, throughly implemented in the paper. I recommend it to be published in the Universe.
Author Response
We thank the reviewer for their favorable appraisal.
Reviewer 2 Report
General comments The probabilistic formulation of the problem is vague / imprecise and should be expressed much clearer and coherently in formal mathematical terms. It would also be appropriate to point out in the conclusions and abstract that the results are conditional on the assumption of the principle of mediocrity. Throughout the text, there is no consistency in the use of symbols for proportionality and similarity / magnitude / estimate. Please review this for clarity. Time dependence in astrophysical quantities due to the historical evolution of the Universe appears to be ignored. Which epochs of the history of the Universe that the work addresses should be made clear, and the relevant time dependence discussed and accounted for where relevant. Since an argument is made for the multiverse, it would seem that an appropriate measure should consider the full evolutionary history of universes populating the multiverse and the number of observer moments during those histories. On a related note, the notion of "the number of observers a universe produces" appears ambiguously defined, as the authors admit. This could be ok, but here it is vague in the sense that it is not specified during which time interval this is considered. It is stated in the introduction that assumptions about having terrestrial planets are not made, yet several of the derivations of origin of life probabilities rely on the terrestrial planet case. It is not clear to what extent this treatment of the problem is self-consistent. It appears that a background assumption in this work is that the mathematical form of the laws of nature remains constant throughout the multiverse. This is an important assumption which limits the type / level of multiverse that is considered, and should be explained in the text. Section 1 It is claimed that p_prior(x) is relatively unimportant and relatively well-understood. It is not demonstrated that this is an established fact, or otherwise likely to be true. But further to the point, the authors have not demonstrated that for *their* results and conclusions, this factor is unimportant. Section 2 Eq. (12) is used with k = 1. Since the impact of the physical constants can potentially vary significantly given a relatively small change in k, it would be relevant to see a discussion of this issue -- motivation and / or sensitivity analysis. Section 5 Eq. (36)-(38) and the following appears to suggest that the number of galaxies with different mass does not change when the gravitational constant is varied. The authors should clarify this point. Section 6 It ought to be discussed how the new habitability factors introduced relate (or not) to and are consistent with the origin of life scenarios now presented. Three extra factors besides the five physical constants are included in the analysis. These are not clearly introduced and described in the text. How are they computed? How are they related (or not) to the physical constants? As relevant, which point in time do they refer to? It is not clear which of the multiple included factors drive the results in Fig 1, 2 and Table 2. A much more thorough discussion of this should be included.Author Response
Please see the attachment.
